# Recycled Eco-Concretes Containing Fine and/or Coarse Concrete Aggregates. Mechanical Performance

Pablo Plaza [1,*] , Isabel Fuencisla Sáez del Bosque [1], Javier Sánchez [2] and César Medina [2,*]

1    School of Engineering, University of Extremadura, UEx-CSIC Partnering Unit, Institute for Sustainable Regional Development (INTERRA), 10003 Cáceres, Spain; isaezdelu@unex.es
2    Eduardo Torroja Institute for Construction Science, Spanish National Research Council (CSIC), 28033 Madrid, Spain; javier.sanchez@csic.es
*    Correspondence: pablopc@unex.es (P.P.); cmedinam@unex.es (C.M.)

**Abstract:** This study analysed the effect of substituting different percentages of natural aggregate with recycled aggregate from concrete crushing, using a coarse fraction as well as a fine fraction. Natural and recycled materials were classified in order to analyse the mechanical performance and impermeability of these eco-concretes in the fresh state as well as in the hardened state. A statistical analysis also determined whether the performance loss was significant from a statistical point of view, finding strength decreases of less than 13% in compressive strength and losses of less than 20% in flexural strength. An increasing trend was found in permeability as the percentage of recycled aggregate in the mix increased.

**Keywords:** mechanical performance; coarse and fine recycled aggregates; permeability; fresh-state concrete properties

## 1. Introduction

The construction sector has a high demand for natural resources and is one of the activities that generates the largest amount of waste during all phases of the construction process (construction, maintenance and demolition). Throughout the EU specifically, the amount of waste generated in the construction sector has been increasing in recent years. According to Eurostat [1], a total of 6.81 Gt of waste was generated in the period from 2004–2020 in the construction sector in the EU-27 as a whole, increasing its magnitude year by year, rising from 29.66% of total waste in 2004 to 37.11% in 2020.

The construction sector also has high $CO_2$ emissions in the extraction, raw material manufacturing and transport processes and high energy consumption. It is estimated that the construction sector is globally responsible for 33% of annual $CO_2$ emissions [2] and 40% of global energy consumption [3]. The high energy consumption and emissions are due to manufacturing concrete components, especially cement, which accounts for 73% of total emissions in the sector [4], and transporting these components.

In this context, the use of recycled materials can be a great advantage in logistical terms, since the distance of transporting raw materials is reduced by having construction and demolition waste territory that can be converted into recycled aggregates available throughout the [5], especially in areas where natural aggregate is scarce and/or impossible to extract [6]. The impact on the environment is also significantly reduced by reducing the exploitation of natural resources in quarries as well as waste deposits in landfills [7]. As a whole, the use of coarse recycled aggregate can reduce greenhouse gas emissions by up to 65% [8], a percentage that could increase even further if fine aggregate is added, which would be advantageous, since sand consumption worldwide is increasing year by year, with an estimate of 47.5 billion tonnes by 2023 [9] and up to 60 billion tonnes by 2030 [10].

However, the main problem in the use of recycled aggregates for manufacturing concrete is limitations at the regulatory level, which restrict the use of recycled aggregates,

in most cases enabling substitution percentages of less than 60% for the coarse fraction of recycled aggregate from concrete crushing. Table 1 shows the amount of recycled concrete aggregate permitted within the different international regulations, indicating the permitted granulometric fraction (coarse or fine), maximum substitution percentage and maximum strength class.

**Table 1.** Regulatory framework for the use of recycled aggregates in concrete manufacture.

| Country | Aggregate Type | Fraction | Max. Substitution (%) | Concrete Type | Strength Class |
|---|---|---|---|---|---|
| Australia AS 1141.62/HB 155:2002 [11] | RCA (Class 1A) | Coarse | 30 | Structural | C40/50 |
| China GB/T-25177 [12] | RCA—Type I | Coarse | 100 | Structural | No limit |
| | RCA—Type II | | 30 | Structural | C40/50 |
| | RCA—Type III | | 30 | Structural | C25/30 |
| | RCA—Type I | Fine | 100 | Structural | C40/50 |
| | RCA—Type II | | 30 | Structural | C25/30 |
| | RCA—Type III | | 30 | Non structural | - |
| Korea KS-F-2573 [13] | RCA | Coarse | 30 | Structural | 27 MPa |
| | | Coarse + Fine | 30 | Non structural | 21 MPa |
| Hong Kong CS-3:2013/HKBD 2009/WBTC-No. 12 [14] | RCA | Coarse | 20 | Structural | C25/30–C35/45 |
| | | | 100 | Non structural | |
| Japan JIS-5021 [15]/JIS-5022 [16]/ JIS-5023 [17] | RCA—HQ | Coarse | 100 | Structural | C45/55 |
| | | Fine | 100 | | |
| | RCA—MQ | Coarse | 100 | Structural | C35/45 |
| | | Fine | 100 | | |
| | RCA—LQ | Coarse | No limit | Non structural | - |
| | | Fine | | | |
| Belgium PTV 406-2003 [18]/NBN B 15-001 [19] | RCA—Type A | Coarse | 50, 30, 20 | Structural | C30/37 |
| Germany DIN 4226-101, DAfStb [20] | RCA—Type A | Coarse | 45, 35, 25 | Structural | C30/37 |
| Italy NTC-2008 [21] | RCA 1 | Coarse | 30 | Structural | C30/37 |
| | | | 60 | | C25/30 |
| | RCA 2 | | 15 | | C45/55 |
| Denmark DS 2426/DCA No. 34 [22] | RCA 1 | Coarse | 100 | Structural | C40/50 |
| | RCA 2 | Coarse and fine | 100 | | |
| Netherlands NEN-5905 [23] | RCA | Coarse | 20 | Structural | C55/67 |
| Portugal LNEC-E471 [24] | RCA 1 | Coarse | 25 | Structural | C40/50 |
| Switzerland MB-2030 [25] | RCA 1 | Coarse | 100 | Structural | No limit |
| | RCA 2 | | 100 | | |

**Table 1.** *Cont.*

| Country | Aggregate Type | Fraction | Max. Substitution (%) | Concrete Type | Strength Class |
|---|---|---|---|---|---|
| United Kingdom BS 8500-2 [26] | RCA | Coarse/Fine | 20 | Structural | C40/50 |
| France NF P 18-545 [27] | RCA 1 | Coarse | 60, 30, 20 | Structural | No limit |
| | RCA 2 | | 40, 15 | | |
| Spain Structural Code [28] | RCA | Coarse | 20 | Structural | C40/50 |
| | | | 100 | Non structural | - |
| EN 206 [29] | RCA | Coarse | 50, 30 | Structural | No limit |
| RILEM | RCA | Coarse | 100 | Structural | C50/60 |
| Brazil NBR 15116 [30] | RCA | Coarse/Fine | 100 | Non structural | - |

There are numerous works that have studied the use of recycled aggregates from concrete crushing in structural concrete manufacture. Jayasuriya [31] statistically analysed a large database with experimental results of concretes that included a coarse fraction of recycled concrete aggregates, reaching the following conclusions: (i) the optimal substitution percentage is below 20%, where the best properties in concrete are obtained; (ii) although strength losses occur as the substitution percentage and the use of homogeneous aggregates increases (as in the case of total substitution), it provides better results as the entire fraction has the same properties; (iii) the strength of concrete with recycled aggregates is affected by an increase in the effective water/cement ratio; and (iv) fracture behaviour is unpredictable due to the notable differences in rigidity within the concrete matrix, on which point further study is required.

Several authors also conclude that recycled concrete aggregates can also be used regardless of the fraction and substitution percentage if optimal quality of the new concrete is achieved. Etxeberría [32] maintains that recycled coarse aggregate can be used for concrete in medium–low-strength concrete (20–45 MPa) even if strength variations of up to 25% are recorded for total substitution of natural coarse aggregate by concrete aggregate. Limiting strength makes it possible to avoid an increase in the amount of cement, which would be counter-productive from an economic and environmental point of view. Along the same lines, McNeil [33] maintains that although there are differences in behaviour, the requirements specified in structural standards are met in real structures, which means that recycled aggregates can be used in structural concrete even if the quality is slightly lower. If the fine fraction is also substituted with recycled concrete sand, the strength reduction is often greater, since the fine fraction has greater influence on strength than the coarse fraction. Tang [34] analysed the properties of concrete with 100% recycled aggregates, observing losses of up to 28.6% for total substitution of both fractions. Regarding other properties, the elastic modulus and tensile strength have similar behaviour, with losses of between 20% and 33% depending on the amount of recycled aggregate and the fraction chosen. Along the same lines, Kenai [35] observed decreases in all the mechanical properties of concrete using recycled aggregates until reaching total substitution. The decrease in compressive strength, of approximately 50% in concrete completely composed of recycled aggregates, presents a considerably greater water demand, mainly because the recycled aggregate has greater absorption. This behaviour can be explained by analysing the micro-structure of the concrete. Several authors [36,37] maintain that the performance loss of concrete with recycled aggregate is due to lower-strength ITZ between the mortar layer adhering to the recycled aggregate and the mortar that is formed while mixing the concrete. The cracks and pores present in this layer of old mortar make it the weakest point of the new concrete,

which also has a greater demand for water due to the increased absorption of this layer of mortar [38].

The variability of results mentioned above lies in the difference in the quality of the recycled aggregates depending on the concrete of origin. Kumar [39] observed that the use of aggregates from medium–low-strength (30 MPa) but high-quality concrete did not affect the strength for substitution percentages of up to 20%. On the contrary, substituting the coarse and fine fractions, separately as well as together, obtained slightly higher strengths with small adjustments in the dosage, obtaining a high-performance concrete (HPC). Other authors have also observed recycled concrete that is more resistant than the reference mixes in the long term, using a coarse fraction to completely substitute the natural aggregate [40] as well as both fractions, substituting the fine fraction in lower percentages [41], defining the optimal percentage below 60% [42].

This work attempts to deepen the study of the physical/mechanical behaviour of concrete that simultaneously includes coarse and fine aggregate from concrete crushing. Specifically, this work analyses the effect of partially (25, 50 and 75%) or totally (100%) substituting the coarse fraction of the natural aggregate with recycled aggregate and at the same time partially substituting (10, 20 and 50%) the fine fraction of the aggregate in the design of class C30/37 structural concretes (characteristic strength of 30 N/mm$^2$). The properties in the fresh state (density, consistency and entrained air) and hardened state (compressive and flexural strength and water penetration under pressure) of the concrete have thus been studied. These results have been analysed from a statistical point of view using two techniques: in the case of density, a linear regression model was studied based on the total percentage of substituted aggregate and in the rest of the properties, an analysis of the variance (ANOVA) with two factors and interaction, which analyses the relative effects and interferences of substituting the coarse and fine fractions separately as well as simultaneously, indicating which changes in the properties studied are significant from a statistical point of view.

## 2. Materials and Methods

### 2.1. Materials

The natural aggregate used to manufacture the concrete comes from crushing greywacke (Figure 1). It has an irregular shape and marked edges and is supplied in three granulometric fractions: 0/6 mm (finely crushed stone, CS-F), 6/12 mm (medially crushed stone, CS-M) and 12/20 mm (coarsely crushed stone, NG-C). Regarding its chemical composition, it is a siliceous aggregate with around 60% $SiO_2$, as well as other oxides in a smaller proportion ($Al_2O_3$, $Fe_2O_3$, MgO and $Na_2O$). From the mineralogical point of view it is characterised by having quartz, feldspars and phyllosilicates.

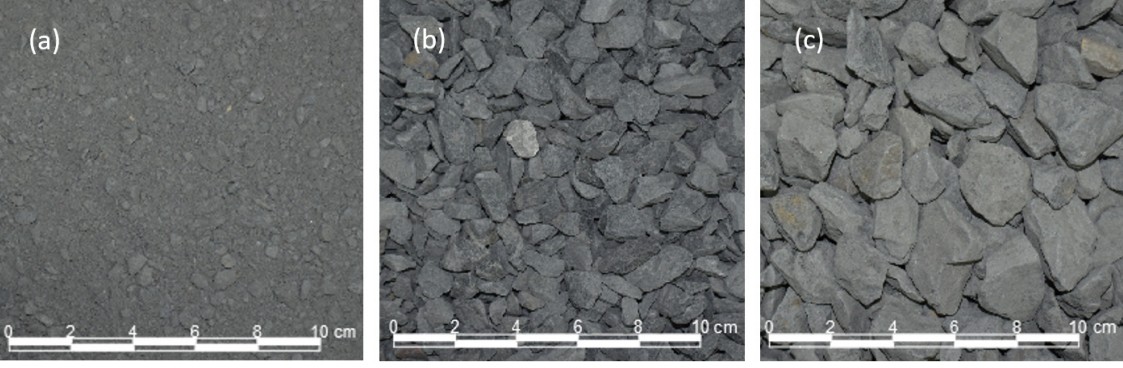

**Figure 1.** (**a**) Finely crushed stone (CS-F); (**b**) medially crushed stone (CS-M); (**c**) coarsely crushed stone (CS-C).

Also, the recycled aggregate comes exclusively from crushing concrete (Figure 2). As with natural aggregates, it is supplied in three granulometric fractions: 0/6 mm (recycled

concrete sand, RCF), 6/12 mm (recycled crushed gravel, RCG) and 12/20 mm (recycled crushed concrete, RCC).

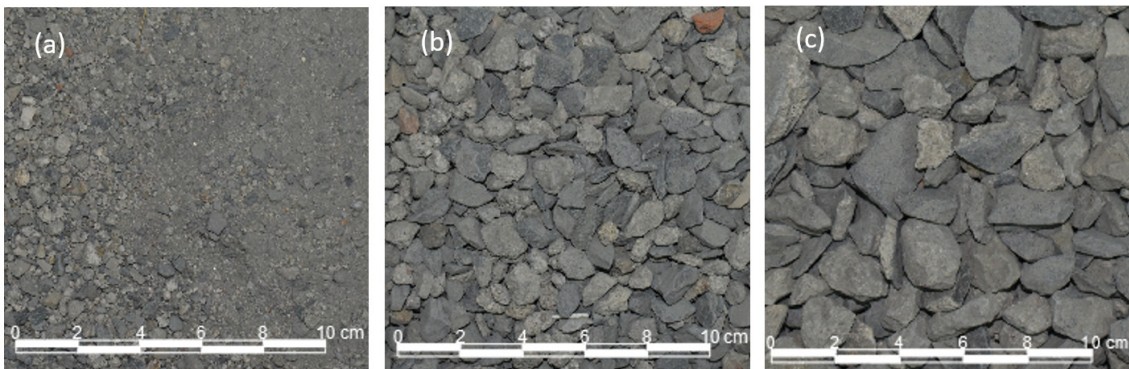

**Figure 2.** (**a**) Recycled Concrete Sand (RCS); (**b**) Recycled Concrete Gravel (RCG) and (**c**) Recycled Crushed Croncrete (RCC).

All aggregates were supplied by ARAPLASA, an aggregate recycling plant located in Plasencia, north of the province of Cáceres (Spain).

The Portland cement used was a CEM I 42.5 R supplied by the Lafarge Holcim plant located in Villaluenga de la Sagra, in the Spanish province of Toledo. This cement meets all the requirements of the EN-197 standard [43].

Finally, the super-plasticising additive FUCHS BRYTEN NF supplied by FUCHS Lubricantes S.A.U. (Bacerlona, Spain) was used, which consists of a modified water-based polycarboxylate. This brown additive is free from chlorides and has a density of 1.1 $g/cm^3$, pH = 8.0 and 20% solids content.

### 2.2. Aggregate Characterisation

Figure 3 shows the composition of gravel and fine gravel recycled from concrete, respectively. It shows that regardless of the recycled coarse fraction, the content of concrete or mortar (Rc) and unbound aggregates (Ru) is ≥95% by weight. Based on this result and according to the classification proposed by the Structural Code (CodE), they can be classified as aggregates from concrete crushing (Rc+Ru ≥ 95%).

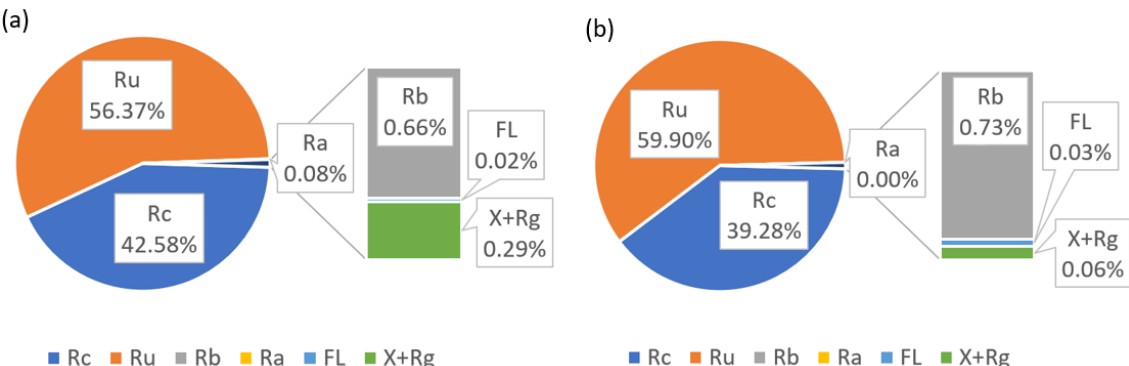

**Figure 3.** Classification for the constituents of coarse recycled aggregate. (**a**) RCG, (**b**) RCC (EN 933-11) [44].

Table 2 shows the physical, chemical and mechanical properties of the aggregates used in the formulation of concrete, as well as the EN 12620 standard requirements [45].

**Table 2.** Physical, chemical and mechanical properties of the aggregates.

| Property [Standard] | CS-F | RCF | CS-M | CS-C | RCG | RCC | EN 12620 |
|---|---|---|---|---|---|---|---|
| Density (Mg/m$^3$) [46] | 2.82 | 2.79 | 2.78 | 2.77 | 2.72 | 2.73 | - |
| Sorptivity (wt%) [46] | 1.18 | 4.42 | 0.88 | 0.78 | 5.40 | 3.63 | <5 |
| Fine equivalent (wt%) [47] | 73 | 61 | - | - | - | - | >70 * |
| LA coefficient (wt%) [48] | - | - | 16 | 18 | 27 | 27 | ≤40 |
| Flakiness index (wt%) [49] | - | - | 20.36 | 24.79 | 16.08 | 20.85 | <35 |
| Water-soluble chlorides (wt%) [50] | <0.01 | | | | | | <0.05 |
| Acid soluble sulphates (wt%) [50] | <0.002 | | | | | | <0.80 |
| Total sulphates (wt%) [50] | <0.001 | | | | | | <1 |

* Aggregates intended for concrete elements exposed to exposure class X0 or XC. Note: CS-F: natural sand, RCF: recycled sand, CS-M: natural gravel—medium, CS-C: natural gravel—coarse, RCG: recycled gravel—medium, RCC: recycled gravel—coarse.

In terms of density, the values of the recycled aggregates are lower than those of natural aggregates, with a decrease of 1.44%, 2.16% and 1.06% for the coarse aggregate (RCC), the medium (RCG) and the fine (RCF), respectively. This decrease is mainly due to the adhered mortar layer present in this type of recycled aggregate, which is less dense and more porous than natural aggregate. The values obtained are similar to those observed by Andreu [51] and Gao [52], who observed values of 5.21% and 4.50%, respectively, for recycled concrete aggregates.

Water absorption of recycled aggregates is between 3.5 and 6 times greater than that of natural aggregates depending on the fraction, coarse or fine, analysed. In all cases however, the provisions of the Structural Code are complied with, which limits the absorption value to 7% for recycled aggregates and 4.5% for natural aggregates. Additionally, the granular skeleton has a water absorption of less than 5% by weight, the maximum limit imposed by the EN 12620 standard. The values obtained are similar to those reported by other authors [51,53,54], who observed values of 3.74% and 5.91%, respectively, for all the recycled concrete aggregate fractions studied.

Regarding the values obtained from the Los Angeles (LA) coefficient, the recycled coarse fractions have a value slightly higher proportion (27% by weight) than that corresponding to the natural fraction (16–18%) due especially to the lower resistance to fragmentation of the adhered mortar, which is more friable than natural aggregates. However, the recorded values are below the 40% by weight (LA40) and 50% by weight (LA50) required for recycled concrete crushing aggregates by the CodE and recycled concrete aggregates type A and B pursuant to the EN 206 standard, respectively. The values obtained are also similar [53], although Andreu [51] observed very low values (10–16%) analysing recycled aggregates from high-strength concrete.

Regarding chemical properties, it should be noted that all aggregates comply with the Structural Code limitations regarding the content of organic matter, water-soluble chlorides and sulphates, including total as well as acid-soluble.

Regarding geometric properties, the flakiness index of recycled aggregates is lower than that of natural aggregates, due to its more rounded shape and less sharp edges (see Figure 2) associated with the construction and demolition waste crushing process and the presence once again of adhered mortar. These values are below the 35% required in the CodE and are similar to those found by other authors [51,53]. Additionally, regarding the quality of the fines, the sand equivalent of the recycled aggregate (RCF) is lower than the minimum required by the CodE for the 0/4 fraction used in concrete elements exposed to exposure classes X0 and XC.

Figure 4 shows the granulometric distribution of the aggregates used, as well as the upper, lower and fine limit content (<0.063 mm) established in the CodE for the fine fraction, observing that all granulometric fractions 0/6, 6/12 and 12/22, regardless of their nature (natural or recycled), have a similar granulometric distribution.

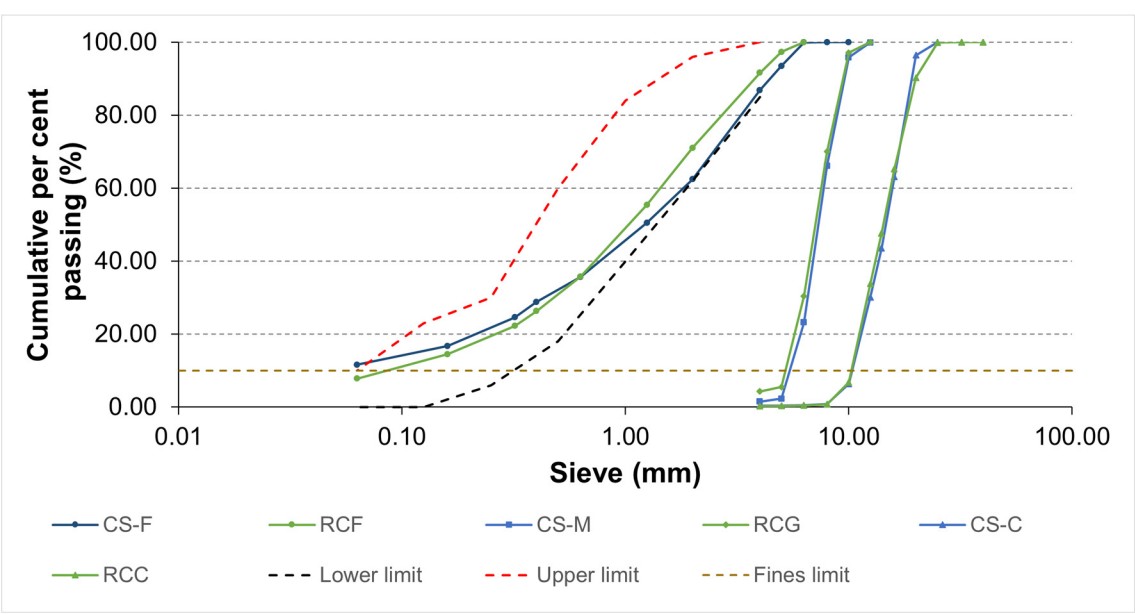

**Figure 4.** Aggregate particle size distribution.

Finally, the fine content (<0.063 mm) of the natural aggregate (CS-F) is 11.59%, slightly over the limit established for concrete with non-limestone aggregates subject to general exposure classes X0 or XC and not subject to any specific XA, XF or XM exposure class. In the case of recycled sand (RCF), the fine content is 7.78%.

### 2.3. Mix Design

Table 3 shows the composition of the 20 formulated mixes: (i) 1 reference mix (HP) with 100% natural aggregate; (ii) 4 mixes with recycled coarse aggregate in different percentages (25%, 50%, 75% and 100%) and 0% recycled sand (HR-25, HR-50, HR-75 and HR-100); (iii) 5 mixes with recycled coarse aggregate in different percentages (0%, 25%, 50%, 75% and 100%) and 10% recycled sand (HR-0+10, HR-25+10, HR-50+10, HR-75+10 and HR-100+10); (iv) 5 mixes with recycled coarse aggregate in different percentages (0%, 25%, 50%, 75% and 100%) and 20% recycled sand (HR-0+20, HR-25+20, HR-50+20, HR-75+20 and HR-100+20); and (v) 5 mixes with recycled coarse aggregate in different percentages (0%, 25%, 50%, 75% and 100%) and 50% recycled sand (HR-0+50, HR-25+50, HR-50+50, HR-75+50 and HR-100+50).

The starting data required to design the mixes according to the DOE British Method [55] were: (i) characteristic design strength ($f_{ck}$) of 30 MPa (C30/37); (ii) effective water-to-cement ratio (w/c) of 0.45; and (iii) 20 mm maximum size of the coarse aggregate. It is also considered that the aggregates are dry, and 70% of the water absorption of the recycled aggregates has been added to the theoretical water content resulting from the dosing process, thus guaranteeing that all mixes have the same amount of water available for cement hydration regardless of the aggregate mix. For manufacturing all the mixes, a super-plasticising additive (6.20 kg/m³) was also added with an amount of 1.55% by weight of cement.

Finally, all mixes meet the minimum dosage requirements (maximum water/cement ratio and minimum cement content) specified in article 43.2.1 of the CodE for use as structural concrete for exposure classes XC1/XC2 and XC3/XC4 with maximum water/cement ratio of 0.60 and 0.55, respectively. Regarding the minimum cement content, a value of 275 kg/m³ and 300 kg/m³ of cement is established for classes XC1/XC2 and XC3/XC4, respectively.

**Table 3.** Mix batching.

| Mix | Components (kg/m$^3$) | | | | | | | |
|---|---|---|---|---|---|---|---|---|
| | NS | RS | NG-M | NG-C | RG-M | RG-C | Cement | Water |
| HP | 732.36 | 0.00 | 382.96 | 766.69 | 0.00 | 0.00 | 400.00 | 193.03 |
| HR-0+10 | 655.65 | 70.59 | 380.95 | 762.65 | 0.00 | 0.00 | 400.00 | 195.20 |
| HR-0+20 | 581.26 | 140.81 | 379.94 | 760.63 | 0.00 | 0.00 | 400.00 | 197.40 |
| HR-0+50 | 360.4 | 349.21 | 376.92 | 754.58 | 0.00 | 0.00 | 400.00 | 203.93 |
| HR-25 | 724.65 | 0.00 | 284.20 | 568.96 | 92.01 | 186.72 | 400.00 | 197.37 |
| HR-25+10 | 648.72 | 109.24 | 282.69 | 565.94 | 91.52 | 185.72 | 400.00 | 202.36 |
| HR-25+20 | 576.64 | 218.48 | 282.69 | 565.94 | 91.52 | 185.72 | 400.00 | 206.34 |
| HR-25+50 | 358.47 | 347.35 | 281.18 | 562.91 | 184.73 | 281.18 | 400.00 | 209.38 |
| HR-50 | 716.94 | 0.00 | 187.45 | 375.27 | 182.06 | 369.46 | 400.00 | 203.86 |
| HR-50+10 | 643.51 | 108.37 | 186.95 | 374.26 | 181.57 | 368.47 | 400.00 | 207.72 |
| HR-50+20 | 572.01 | 216.73 | 186.95 | 374.26 | 181.57 | 368.47 | 400.00 | 211.67 |
| HR-50+50 | 354.62 | 343.61 | 185.43 | 371.24 | 180.10 | 365.49 | 400.00 | 214.55 |
| HR-75 | 711.16 | 0.00 | 92.97 | 186.12 | 270.88 | 549.72 | 400.00 | 209.18 |
| HR-75+10 | 638.31 | 68.72 | 92.72 | 185.62 | 270.15 | 548.23 | 400.00 | 211.29 |
| HR-75+20 | 567.38 | 137.44 | 92.72 | 185.62 | 270.15 | 548.23 | 400.00 | 213.49 |
| HR-75+50 | 349.80 | 338.94 | 91.46 | 183.10 | 266.48 | 540.79 | 400.00 | 219.48 |
| HR-100 | 701.52 | 0.00 | 0.00 | 0.00 | 356.29 | 723.03 | 400.00 | 214.20 |
| HR-100+10 | 627.90 | 67.60 | 0.00 | 0.00 | 356.29 | 723.03 | 400.00 | 216.15 |
| HR-100+20 | 556.59 | 134.83 | 0.00 | 0.00 | 353.35 | 717.07 | 400.00 | 218.19 |
| HR-100+50 | 345.94 | 335.21 | 0.00 | 0.00 | 351.39 | 713.10 | 400.00 | 224.40 |

*2.4. Experimental*

Prior to the design of the different mixes, the physical, mechanical and chemical properties (see point 2.2) of the aggregates used in this research were analysed. Next, the concrete mixes previously described were theoretically formulated (see Section 2.3) and adjusted on a laboratory scale. The properties of the manufactured concrete (see Table 4) were then studied in their fresh state (density, entrained air and consistency) and in their hardened state (density, compressive and flexural strength and water penetration under pressure) (see Table 4).

**Table 4.** Concrete properties studied.

| Property | Standard | Sample Size (cm) | NS/M | Testing Age (Days) |
|---|---|---|---|---|
| Fresh state | | | | |
| Density | EN 12350-6 [56] | | | |
| Entrained air | EN 12350-7 [57] | Evaluated during the manufacturing process | | |
| Consistency | EN 12350-2 [58] | | | |
| Hardened state | | | | |
| Density | EN 12390-7 [59] | 15 × 15 × 15 | 3 | 28 |
| Compressive strength | EN 12390-3 [60] | 15 × 15 × 15 | 9 | 7, 28, 90 |
| Flexural strength | EN 12390-5 [61] | 10 × 10 × 40 | 3 | 28 |
| Water penetration under pressure | EN 12390-8 [62] | Ø15 × 30 | 3 | 28 |

Note: NS/M: number of samples/mix.

Figure 5 shows the procedure followed for mixing the designed mixes, manufacturing and subsequent curing of the samples. The first phase of the mixing process consists of loading and homogenising the materials. To do so, the granular skeleton consisting of the aggregates was first placed into the mixer and then mixed for 30 s. The cement was

then added, mixing for another 60 s. The second phase consists of the mixing itself, which begins by diluting the additive in 10% of the mixing water. This mix is added to the mixer for 45 s, after which 70% of the mixing water is added. Finally, the remaining 20% is added, and everything is mixed for 240 s. Finally, the third phase consists of filling the moulds and compacting them by chopping with a bar as established by the EN 12390-1 standard [63].

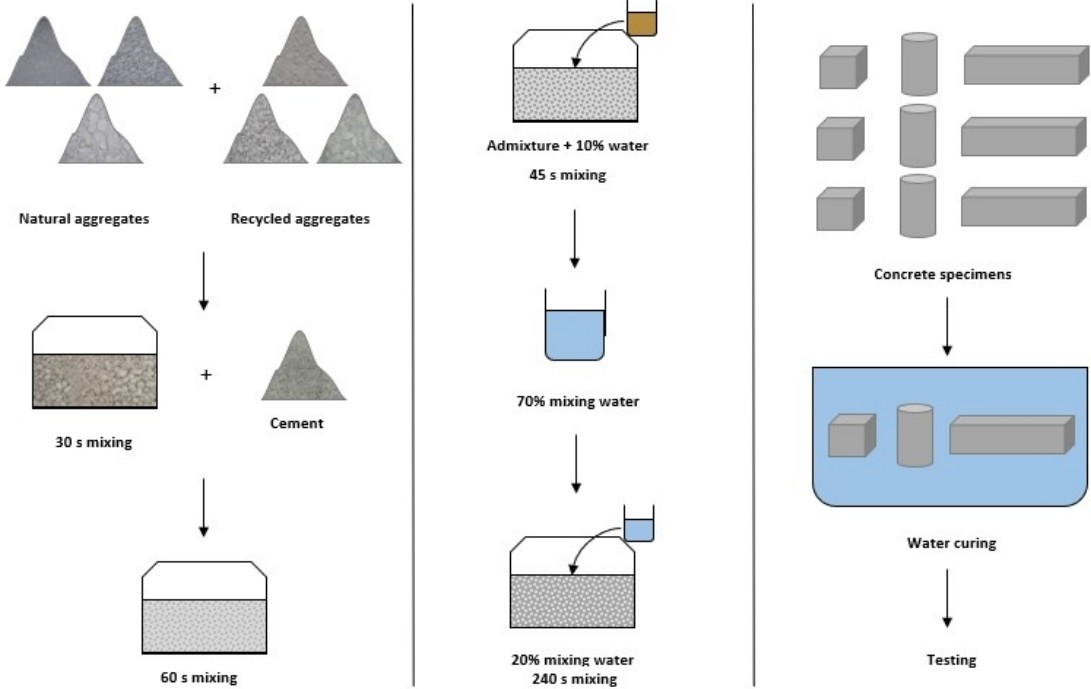

**Figure 5.** Mixing process.

*2.5. Statistical Analysis*

To evaluate the influence of replacing natural aggregate with recycled aggregate on concrete properties, two different techniques were used, depending on the property studied. First, to study the mechanical properties of the concrete, an analysis of variance (ANOVA) was carried out for each property and age studied, carrying out a total of four analyses. The statistical software "R", version 4.0.5, was used to make the calculations.

The proposed model (Equation (1)) to carry out the analyses corresponds has two factors (% substitution of coarse aggregate and % substitution of fine aggregate) and interactions.

$$Y_{ijk} = \mu_{11} + \alpha_i + \beta_j + \alpha\beta_{ij} + \varepsilon_{ijk}, i = 1, \ldots, 4; j = 2; k = 1, \ldots, 3 \tag{1}$$

This model obtains a value for the response variable $Y_{ijk}$ (strength studied in each case) by adding different values: (i) $\mu_{11}$ is the average value corresponding to the reference mix (HP) in each case; (ii) $\alpha_i$ quantifies the relative effect corresponding to the first factor (% coarse aggregate substitution); (iii) $\beta_j$ quantifies the relative effect corresponding to the second factor (% fine aggregate substitution); (iv) $\alpha\beta_{ij}$ is the relative effect due to the interaction that occurs when simultaneously substituting both fractions; and (v) $\varepsilon_{ijk}$ indicates the perturbation of the model.

To check whether the analyses carried out are valid, the homoscedasticity and normality assumptions must first be checked using the Bartlett and Shapiro–Wilk tests, respectively. Table 5 shows the *p*-values of both tests. As can be seen, both assumptions are met in all analyses, so the analyses are therefore considered valid.

**Table 5.** Homoscedasticity and normality tests.

| Contrast Type | Compressive | | | Flexural | Water under Pressure |
|---|---|---|---|---|---|
| | 7 Days | 28 Days | 90 Days | | |
| Bartlett | 0.456 | 0.803 | 0.381 | 0.605 | 0.075 |
| Shapiro–Wilk | 0.237 | 0.627 | 0.556 | 0.231 | 0.532 |

Once the initial assumptions were tested, the model calculates the model coefficients, indicating whether or not these values are significant. In other words, for what percentages of substitution is strength variation relevant or not from a statistical point of view ($p$-value < 0.05).

In the density study (apparent as well as fresh-state), expressions were also observed that relate the total percentage of substituted aggregate (%) with the density (D) studied in each case. In both cases, the model proposed for the optimal adjustment was obtained using linear regression (Equation (2)), in both cases obtaining very good correlation coefficients ($R^2 > 0.8$):

$$D = a\% + b \tag{2}$$

This model returns a density value through an affine function with parameters including the y-intercept (b), which corresponds to the reference value for the mix without recycled aggregate (RC) and the gradient (a) that adjusts the line to the data obtained through a least-squares fitting.

## 3. Results

### 3.1. Fresh-State Properties

Figure 6 shows the fresh-state density as a function of the percentage of total substitution of natural aggregate by recycled aggregate, observing that the density decreases as the recycled aggregate content increases, recording losses of less than 5% in all cases. This decrease is mainly due to the lower density of the recycled aggregates, as well as the greater amount of entrained air (see Table 6) in the mixes with a greater amount of recycled aggregate. This trend was observed by other authors [64,65] incorporating recycled concrete aggregates in all fractions, with decreases of around 3% for total substitution of coarse aggregate.

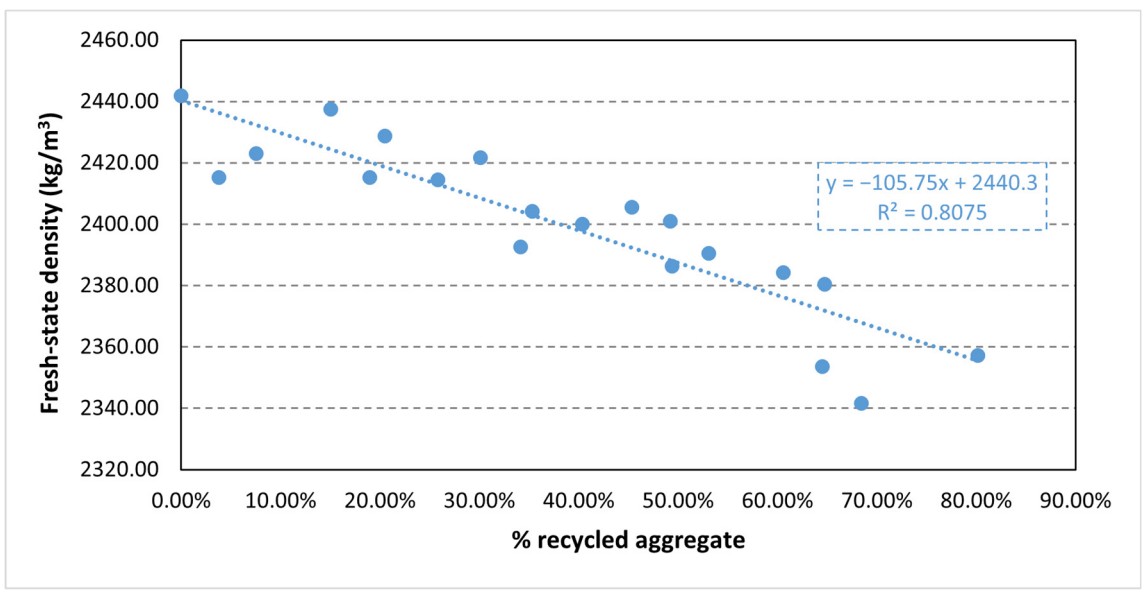

**Figure 6.** Fresh-state density.

**Table 6.** Settlement and entrained air.

| Mix | Slump (mm) | Entrained Air (Vol. %) |
|---|---|---|
| HP | 82.8 | 1.63 |
| HR-0+10 | 65.0 | 1.58 |
| HR-0+20 | 75.0 | 1.60 |
| HR-0+50 | 80.0 | 1.68 |
| HR-25 | 78.0 | 1.66 |
| HR-25+10 | 77.5 | 1.78 |
| HR-25+20 | 77.1 | 1.74 |
| HR-25+50 | 87.0 | 1.80 |
| HR-50 | 89.0 | 1.66 |
| HR-50+10 | 82.0 | 1.66 |
| HR-50+20 | 60.0 | 1.82 |
| HR-50+50 | 77.5 | 1.74 |
| HR-75 | 66.3 | 1.73 |
| HR-75+10 | 90.0 | 1.60 |
| HR-75+20 | 75.0 | 1.73 |
| HR-75+50 | 92.5 | 1.68 |
| HR-100 | 72.0 | 1.82 |
| HR-100F10 | 90.0 | 1.90 |
| HR-100F20 | 90.0 | 1.90 |
| HR-100F50 | 80.0 | 1.90 |

From a statistical point of view, the proposed expression shows a clear linear relationship ($R^2 > 0.8$) between the substitution percentage and the density, enabling the density to be calculated based on the amount of aggregate substituted.

Table 6 shows the slump and entrained air values observed for the different mixes, first showing that all the mixes have a soft consistency (50–90 mm) pursuant to article 33.5 of the CodE, which corresponds to a type S2 settlement according to EN 206. These results confirm that all samples have the same workability regardless of the amount of aggregate substituted, because the amount of super-plasticiser additive as well as the effective water/cement ratio are maintained.

Finally, the entrained air content varies slightly for the different mixes, obtaining values between 1.58% and 1.90%. In this case, the lower density and greater porosity of the recycled aggregates result in a slight increase in air content [66]. Simsek [65] observed a similar behaviour, with slight variations but an increasing trend as the substitution percentage increases, for the coarse fraction as well as for the fine fraction.

### 3.2. Hardened-State Properties

### 3.2.1. Bulk Density

Figure 7 shows the bulk density data as a function of the amount of recycled aggregate of the different mixes, as well as the adjustment made and the proposed mathematical expression. In this case, the proposed expression also shows the linear relationship ($R^2 > 0.8$) between the density and the percentage of recycled aggregate.

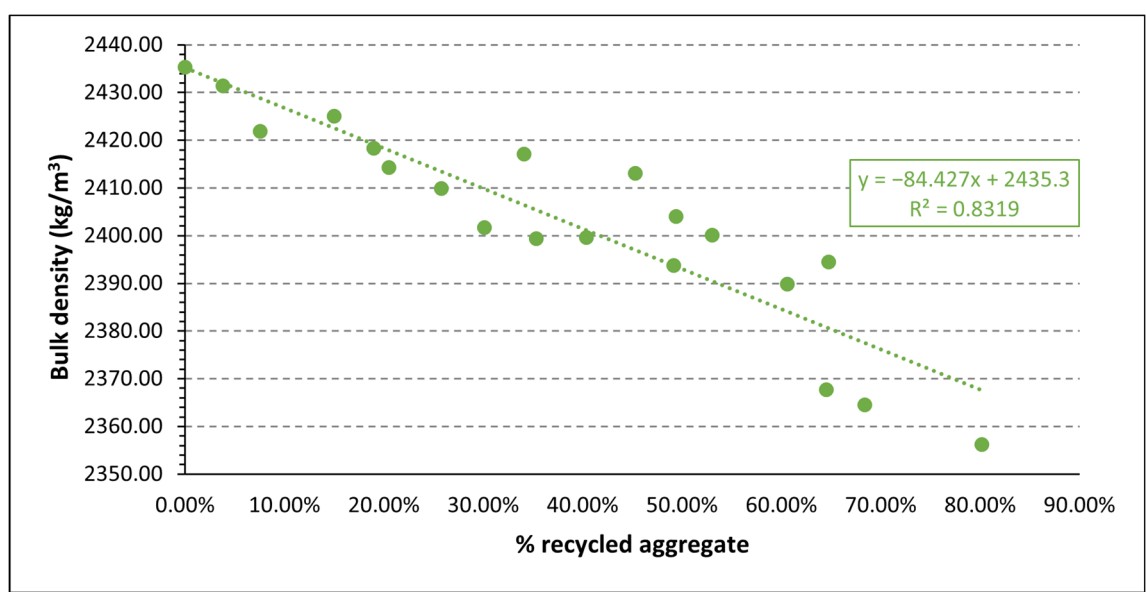

**Figure 7.** Apparent density.

The density behaviour is very similar to that previously described for the fresh-state density, although with slightly lower values. The range of density loss (0.16–3.25%) is lower than that obtained in the fresh state, although the data trend is very similar. Tuyan [67] maintains that this decrease in apparent density is directly related to the presence of macropores in the mortar adhered to recycled concrete aggregates, with a direct relationship between the amount of aggregate substituted and the decrease in density.

### 3.2.2. Compressive Strength

Figure 8 shows the results of compressive strength at age 7, 28 and 90 days in a $150 \times 150 \times 150$ mm cubic sample, observing that all the mixes exceed the corresponding design strength with a C30/37 concrete ($f_{ck} = 30 \ N/mm^2$), which indicates that all mixes, regardless of the content and recycled fraction used, could be used from a point of view of this property in manufacturing structural concrete.

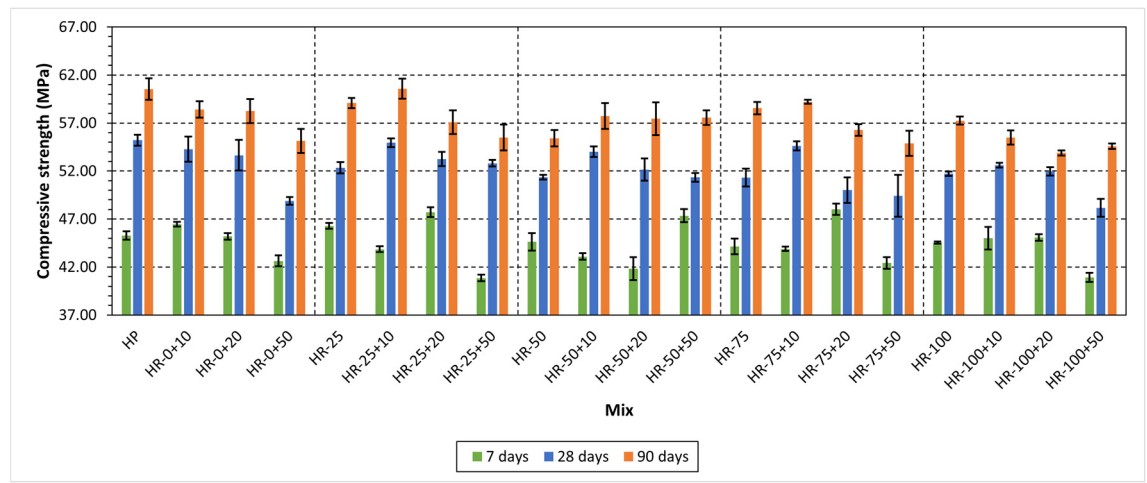

**Figure 8.** Simple compressive strength.

In the case of mixes that include only the coarse fraction of the aggregate, there are small strength losses, which increase slightly as the age increases. At 7 days, the HR-25 mix even presents a higher strength than the reference mix (+2.22%). However, strength

decreases in the rest of the mixes between 1.42% and 2.5%. At 28 and 90 days, the losses are very similar in all mixes, with decreases between 5.20% and 7.04% at 28 days and between 2.39% and 8.45% at 90 days. This behaviour is similar to that found by Chang [68], who observed a 7-day strength increase of 1.35% for a 25% substitution of coarse concrete aggregate. At 28 days, losses of 3.86% and 7% were observed for substitution percentages of 25% and 75%, respectively. Pedro [69] observed losses of between 3.2% and 7.6% for concrete with a target $f_{ck}$ of 45 MPa, substituting 25% to 100% of the coarse fraction with recycled concrete aggregates.

Regarding the fine fraction, the strength loss is similar to that resulting from substituting the coarse fraction for percentages up to 20%. At 7 days, the behaviour is very similar to that described above, even recording a slight increase (2.61%) for the HR-0+10 mix. At 28 and 90 days, the losses range between 1.67% and 5.83%. In the case of the HR-0+50 mix, the strength losses are significantly higher, with losses between 5.83% and 11.46%. This behaviour agrees with the results observed by other authors. Mohammed [70] recorded a decrease in strength of approximately 14% for a 50% substitution of the fine fraction with concrete aggregate, associating these losses with the significant increase in the absorption of the recycled aggregate. For lower substitution percentages, the results observed by Zega [71] are very similar to those found in this work, with losses around 2% substituting 20% of the fine fraction with concrete sand. At 90 days, the difference with respect to the reference concrete is reduced to approximately 1.6%. In this case, a reduction in the effective water/cement ratio leads to an improvement in the interface (ITZ), which results in behaviour very similar to the reference concrete, even improving other durable properties.

Finally, the simultaneous addition of both fractions mitigates the effect of strength loss, obtaining concretes with strength very similar to the previous ones but with a higher recycled aggregate content. The effect of simultaneous addition can be observed with the data from the statistical analysis, whose parameters are shown in Table 7.

From a statistical point of view, the effect of adding coarse aggregate is generally negative, as reflected in the data shown in Figure 9. However, the behaviour at 7 days has non-significant factors that reveal that the trend is not as clear as in the other ages. In the case of fine aggregate, the factors follow the same trend, which corroborates the idea that the addition of any fraction of the aggregate results in a decrease in strength.

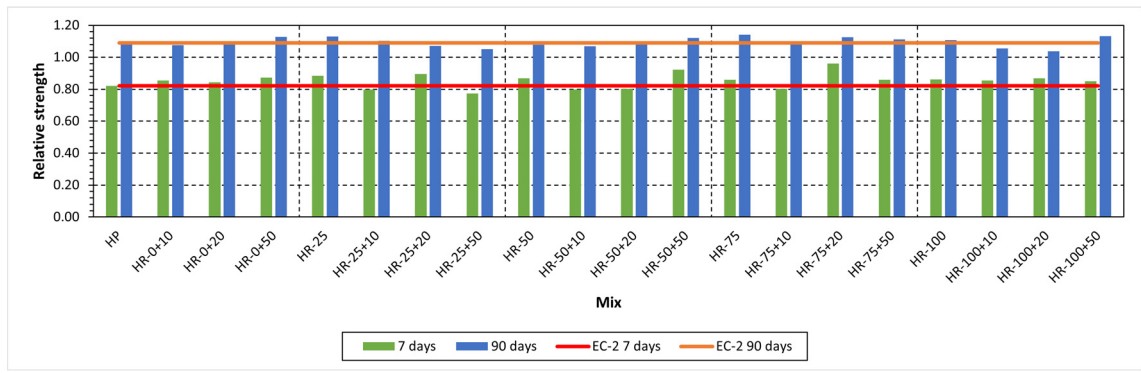

**Figure 9.** Relative strength.

Additionally, the interaction factors reveal positive values for 28 and 90 days in all cases, which indicates that the strength loss is mitigated when both fractions are added simultaneously, which also contributes to obtaining concretes with similar strength to those that have only one fraction type (coarse or fine), although increasing the percentage of recycled aggregate in the mixes.

**Table 7.** Estimation of the parameters of the different models.

| Parameter | Compressive | | | Flexural | Water under Pressure |
|---|---|---|---|---|---|
| | **7 Days** | **28 Days** | **90 Days** | | |
| $\mu_{11}$ | 45.27 | 55.20 | 60.53 | 7.12 | 4.45 |
| $\alpha_2$ | 1.00 | −2.87 | NS | NS | NS |
| $\alpha_3$ | NS | −3.84 | −5.12 | −0.42 | 5.98 |
| $\alpha_4$ | −1.13 | −3.89 | −1.97 | −0.49 | 3.38 |
| $\alpha_5$ | NS | −3.49 | −3.28 | −0.66 | NS |
| $\beta_2$ | 1.18 | NS | −2.12 | NS | −2.51 |
| $\beta_3$ | NS | −1.56 | −2.28 | NS | NS |
| $\beta_4$ | −2.64 | −6.33 | −5.39 | −0.25 | NS |
| $\alpha\beta_{22}$ | −3.60 | 3.53 | 3.62 | | 3.54 |
| $\alpha\beta_{32}$ | −2.72 | 3.57 | 4.44 | | −3.28 |
| $\alpha\beta_{42}$ | −1.44 | 4.24 | 2.78 | | NS |
| $\alpha\beta_{52}$ | NS | NS | NS | | NS |
| $\alpha\beta_{32}$ | 1.51 | 2.47 | NS | | NS |
| $\alpha\beta_{33}$ | −2.71 | 2.36 | 4.32 | No interaction | NS |
| $\alpha\beta_{43}$ | 3.96 | NS | NS | | NS |
| $\alpha\beta_{53}$ | NS | NS | NS | | NS |
| $\alpha\beta_{23}$ | −2.80 | 6.81 | NS | | NS |
| $\alpha\beta_{33}$ | 5.34 | 6.30 | 7.55 | | −4.83 |
| $\alpha\beta_{43}$ | NS | 5.64 | NS | | NS |
| $\alpha\beta_{53}$ | NS | 2.79 | 2.72 | | NS |

Note: NS: not significant.

Regarding the evolution of strength over time, Figure 9 shows the relative strength values of the mixes with respect to the reference value at 28 days, as well as the relative strength estimated in the Eurocode-2 (EC-2) at 7 and 90 days. It shows that all the mixes present a similar evolution in strength gain, in most cases reaching greater strength than expected at 7 and at 90 days. This result shows that the addition of the recycled coarse and/or fine concrete fraction does not influence the cement hydration process. It also indicates that the behaviour recorded at 7 days, in which greater strength is observed than expected by the EC, is concordant with the results of Surendar [72], which evaluated the behaviour of mixes with between 10% and 75% substitution of the coarse fraction with concrete washed aggregate.

Finally, it indicates that all the mixes present the same failure mode regardless of the age studied. This failure mode is classified in the EN 12390-3 standard as satisfactory (Figure 10).

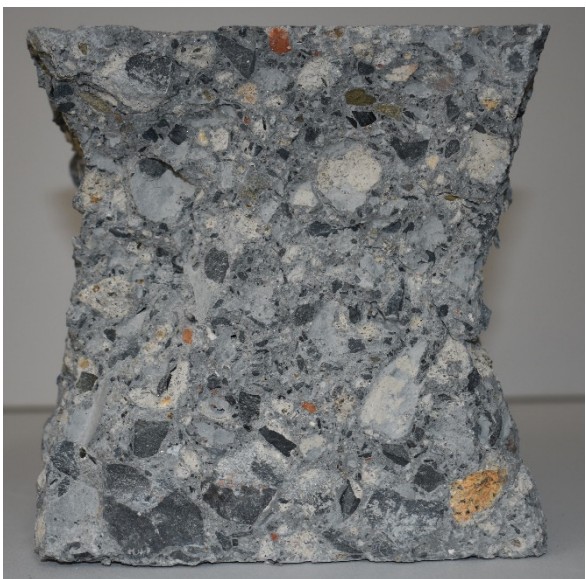

**Figure 10.** Failure mode in HR-100+50 mix.

### 3.2.3. Flexural Strength

Figure 11 shows the flexural strength at 28 days of the formulated concrete, as well as the estimated strength according to the expression included ($f_{ct,m,fl} = 1.6 - h/1000)f_{ct,m}$) in point 3.1.8 of the CodE. Additionally, the flexural strength has been estimated from the compressive strength, using the expression included in article A19.3.1.8, combined with the expression for average tensile strength and taking 90% of the compressive strength obtained in the cubic samples as $f_{ck}$. The resulting expression is shown in the Equation (3)

$$f_{ct,m,fl} = 0.405 \cdot fcm^{2/3} \tag{3}$$

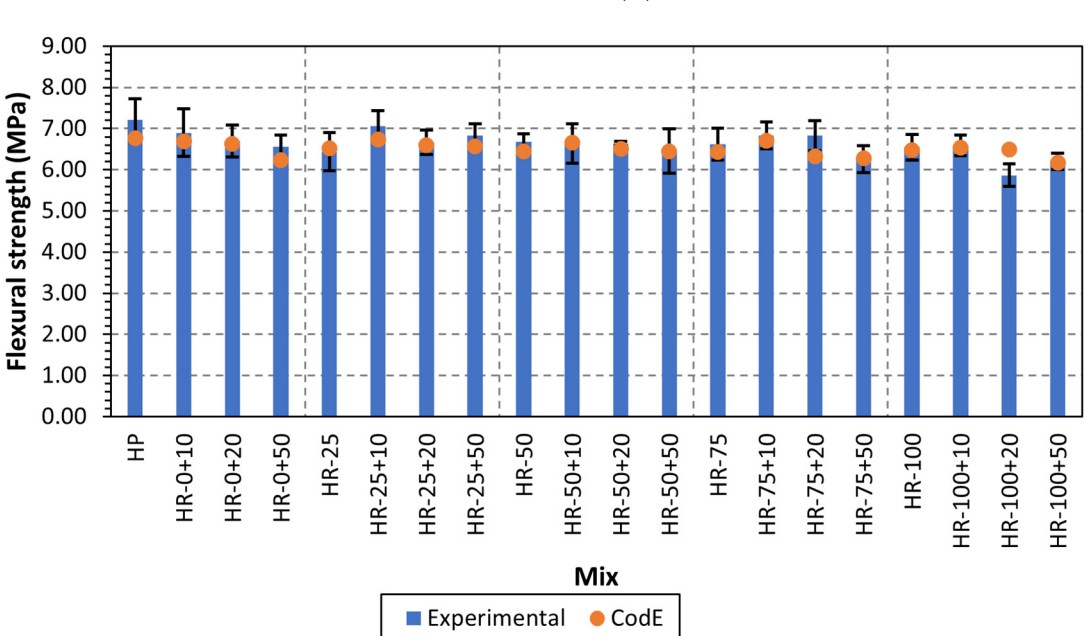

**Figure 11.** Flexural strength.

In the mixes that only substitute the coarse fraction of the aggregate, lower strengths were observed, with losses of between 7.41% and 10.73%, with very similar values regardless of the substitution percentage. In the case of fine aggregate, the decrease in strength is greater as the amount of recycled aggregate increases, with losses of between 4.37% and

9.01%. In the mixes that incorporate both fractions, the values are similar except for mixes with 50% fine aggregate, which generally present a higher strength loss, although all values remain below 20%. These values are similar to those recorded by other authors. Saini [73] and Yaba [74] found losses of 4.1% and 10% for 50% and 100% substitution of the coarse fraction, respectively. Regarding the fine fraction, Mohammed [70] observed a loss of 14% by substituting 50% of the fine aggregate with recycled aggregate.

Mohammed [70] and Xiao [75] state that the reduction in strength occurs due to the presence of micro-cracks in the adhered mortar in the recycled fractions, as well as the intrinsic properties (thickness and micro elastic) of the ITZs old mortar/new mortar and old aggregate/old mortar, whose values are worse than those shown by new aggregate/new mortar.

It should also be noted that in all mixes, regardless of their granular skeleton, the failure mechanism consisted of a single large crack that was initiated in the flexural span and with a normal orientation to the tensile stresses generated due to flexure [76].

From a statistical point of view (Table 7), the effect of adding the coarse aggregate is negative, with a non-significant value for the HR-25 mix. Regarding the mixes that incorporate only the fine fraction, there is only one significant factor for the HR-0+50 mix, so there is no clear trend. Likewise, there is no interaction, so the combination of the coarse and fine fractions has no significant effect on the model studied.

### 3.2.4. Water Penetration under Pressure

Table 8 shows the average and maximum depth of water penetration under pressure, showing that all the mixes, regardless of the percentage and recycled fraction added, have average and maximum depth values below the limits established in the article. 43.3.2 of the CodE ($P_{med} \leq 20$ mm and $P_{max} \leq 30$ mm) for exposure classes (XS3 and XA3). Therefore, all formulated concretes have a sufficiently impermeable structure against water penetration.

**Table 8.** Average and maximum depths of water penetration under pressure.

| Mix | Average Depth (mm) | Maximum Depth (mm) |
| --- | --- | --- |
| HP | 4.45 | 14.54 |
| HR-0+50 | 5.35 | 9.96 |
| HR-25 | 5.25 | 9.09 |
| HR-25+10 | 6.28 | 16.06 |
| HR-25+20 | 5.50 | 12.58 |
| HR-25+50 | 5.81 | 11.43 |
| HR-50 | 9.51 | 14.64 |
| HR-50+10 | 4.64 | 9.48 |
| HR-50+20 | 8.27 | 17.94 |
| HR-50+50 | 6.49 | 11.44 |
| HR-75 | 7.83 | 15.02 |
| HR-75+10 | 5.32 | 10.17 |
| HR-75+20 | 7.53 | 14.78 |
| HR-75+50 | 6.91 | 12.50 |
| HR-100 | 6.21 | 12.97 |
| HR-100+50 | 6.80 | 13.87 |

It firstly indicates that for the mixes that incorporate only the coarse fraction of the recycled aggregate, the average penetration values are higher, presenting high variability in the data obtained, with increases from 17.99% to 113.63%. However, the maximum

penetration values are very similar or even lower in some cases. The fine fraction only has the HR-0+50 mix, which has an average depth 20.15% greater than the reference mix.

In general terms, the variability of the measurements does not allow a clear trend to be established from a statistical point of view, although a slight upward trend can be observed as the recycled aggregate content increases. Analysing the model parameters (Table 7), the substitution of the coarse fraction produces a positive relative effect (increase in the average depth) regardless of the substitution percentage. However, the effect is the opposite for low substitution percentages of the fine fraction (10%). Regarding interferences, there is no clear trend from a statistical point of view, with few significant values.

In general terms, the literature reveals that the depth values increase as the amount of recycled aggregate increases, both coarse and fine [77], although some authors do not consider it significant [78]. Zega [71] observed that the penetration values are slightly higher if the fine fraction of recycled aggregates is used, although the behaviour is very similar for substitution percentages less than 30%. Kapoor [79] recorded depth increases of 30% for total substitution of the coarse fraction with concrete aggregate, as well as increases of 18% when both fractions were combined (100% of the coarse fraction and 50% of the fine). Velardo [80] also observed similar values for average penetration (~8 mm) and somewhat higher values in the case of maximum penetration (~18 mm) using mixed aggregates.

## 4. Conclusions

The conclusions obtained in this work are presented below:

- Recycled aggregates have greater absorption, as well as lower LA coefficient, density and flakiness index than natural aggregates.
- The coarse recycled fractions (gravel and gravel) and fine (sand) comply with the mechanical, physical and chemical requirements set forth in the current regulations on aggregates for concrete.
- The workability of the concrete is not affected by the addition of the recycled fractions (coarse and/or fine), all of which show a soft consistency.
- The density of the concrete with recycled aggregate is lower than that of the reference concrete in all cases, in the fresh state as well as in the hardened state. The density decreases as the proportion of recycled aggregate in the mix increases, registering density variations of less than 5% in all cases.
- The entrained air content increases slightly as the amount of recycled aggregate increases, although remaining within the usual values for conventional reinforced concrete, not exceeding 1.9% in the mixes with the highest recycled aggregate content.
- The compressive strength of concrete with recycled aggregate is lower than that of the reference mix, with losses of less than 13% in all cases. The greatest losses are recorded in mixes that include a higher percentage (50%) of fine recycled aggregate.
- The flexural behaviour is similar to that recorded in compressive, slightly increasing the maximum loss percentage to 19%. Losses are generally greater in mixes that include a high percentage of recycled aggregate, coarse as well as fine.
- All mixes are therefore suitable for use in class C30/37 structural concrete.
- The expression included in the structural code for estimating the flexural strength is correct, showing values with differences of less than 10% compared to the experimental values for all mixes.
- The penetration depths of water under pressure present great variability, with increases of up to 100%. However, the provisions of the regulations are complied with in all cases.

The conclusions indicate that eco-concretes can be used to manufacture structural concrete analyzing mechanical performance. However, it is necessary to complement the tests carried out by analysing the durability of all mixes, as well as on full-scale structural pieces. Furthermore, in order to properly study the environmental benefit, a Life Cycle Analysis (LCA) would be necessary to estimate the environmental benefit taking into account the variables concerning waste treatment or transportation.

**Author Contributions:** Conceptualization, P.P., I.F.S.d.B., J.S. and C.M.; methodology, P.P. and I.F.S.d.B.; software, P.P.; validation, P.P.; formal analysis, P.P.; investigation, P.P. and I.F.S.d.B.; writing—original draft preparation, P.P.; writing—review and editing, P.P., I.F.S.d.B., J.S. and C.M.; supervision, I.F.S.d.B., J.S. and C.M.; project administration, J.S. and C.M.; funding acquisition, C.M. All authors have read and agreed to the published version of the manuscript.

**Funding:** This research was funded by the Spanish Ministry for Science and Innovation under project PDC2022-133285-C21 funded by MCIN/AEI/10.13039/501100011033 and, by the 'European Union NextGenerationEU/PRTR', by Spanish Ministry for Science, Innovation and Universities under project PID2022-136244OB-I00 funded MICIU/AEI/10.13039/501100011033 and by "FEDER/UE" and the IB 20131 research project financed by the Consejería de Economía, Ciencia y Agenda Digital de la Junta de Extremadura and by the European Union Regional Development Fund (ERDF). Author Pablo Plaza benefitted from Spanish Ministry of Education, Culture and Sport pre-doctoral grant FPU19/06704.

**Institutional Review Board Statement:** Not applicable.

**Informed Consent Statement:** Not applicable.

**Data Availability Statement:** The datasets presented in this article are not readily available because the data are part of an ongoing study. Requests to access the datasets should be directed to cmedinam@unex.es.

**Conflicts of Interest:** The authors declare no conflicts of interest.

## List of Abbreviations

| | |
|---|---|
| CDW | Construction and demolition waste |
| RCA | Recycled concrete aggregate |
| CS-F | Natural sand |
| CS-M | Natural gravel—medium |
| CS-C | Natural gravel—coarse |
| RCF | Recycled sand |
| RCG | Recycled gravel—medium |
| RCC | Recycled gravel—coarse |

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
