# Peer review of "Recycled Eco-Concretes Containing Fine and/or Coarse Concrete Aggregates. Mechanical Performance"

_applsci, doi:10.3390/app14103995_

Round 1
Reviewer 1 Report
Comments and Suggestions for Authors
The peer-reviewed paper deals with the issue of replacing different percentages of natural aggregate with recycled aggregate from crushed concrete, with both coarse and fine fractions. Natural and recycled materials were classified in order to analyse the mechanical properties and impermeability of these eco-concretes in the fresh and hardened state. Statistical analysis was used to determine if the loss of mechanical properties was statistically significant, with strength decreases of less than 13% for compressive strength and less than 20% loss for flexural strength. An increasing trend was found for permeability as the percentage of recycled aggregate in the mix increased.
The paper has a classic structure in its composition. The title of the paper is apt, as is the abstract of the paper. The text of the paper is divided into five basic parts. The individual parts of the paper are, in my opinion, well balanced. The English used is understandable and error-free. The attached tables, images and graphs are clear and appropriately complement the text of the presented issue. The equations used are given in correct form without errors. The procedures designed and used are applied correctly. The large number of references to the literature used should be appreciated. The presented conclusions are in line with trends in the subject area and, in my opinion, are correct. The obtained results enrich the issue and may be of interest to the technical public. Even if the contribution follows on from previous works and their authors, in my opinion it can be considered original with original outputs. Overall, I can sum up that this is a well-written paper suitable for publication. Nevertheless, I recommend that the authors consider supplementing the text of the paper with the following:
· For which area, for which structures will eco-concrete be suitable?
· I recommend supplementing the conclusion with the idea of using other types of tests within the framework of real test samples (focused, for example, on the shrinkage process, other mechanical, load tests, on the formation of cracks, etc. Let us point out that within the scope of the issue in question, it may be appropriate to also focus on the problem of fire resistance etc.
· I also recommend supplementing the conclusion of the contribution with information on how to develop new technological procedures in the given issue, what are the trends and application possibilities of these mixtures, etc.
· Provide at least an estimate of the percentage environmental benefit.
· I recommend indicating the direction in which the research and development will go, or indicating the necessary follow-up work.
Author Response
The responses to the comments made by you after reviewing the paper can be found below:
- For which area, for which structures will eco-concrete be suitable?
The conclusions of the paper indicate that all mixtures can be used for structural concrete of class C30/37.
- I recommend supplementing the conclusion with the idea of using other types of tests within the framework of real test samples (focused, for example, on the shrinkage process, other mechanical, load tests, on the formation of cracks, etc. Let us point out that within the scope of the issue in question, it may be appropriate to also focus on the problem of fire resistance etc.
- I also recommend supplementing the conclusion of the contribution with information on how to develop new technological procedures in the given issue, what are the trends and application possibilities of these mixtures, etc.
- Provide at least an estimate of the percentage environmental benefit.
- I recommend indicating the direction in which the research and development will go, or indicating the necessary follow-up work.
In response to these recommendations, the following paragraph has been added to point 5 (conclusions):
"The conclusions indicate that eco-concretes can be used to manufacture structural concrete analyzing mechanical performance. However, it is necessary to complement the tests carried out by analysing the durability of all mixes, as well as on full-scale structural pieces. Furthermore, in order to properly study the environmental benefit, a Life Cycle Analysis (LCA) would be necessary to estimate the environmental benefit taking into account the variables concerning waste treatment or transportation."
Reviewer 2 Report
Comments and Suggestions for Authors
The article is detailed. The state of the problem is analyzed in detail and detailed studies are presented on the influence of recycled aggregates on the properties of concrete.
Notes:
1. The type of rock from which the recycled aggregates are obtained is unknown.
2. The strength characteristics of crushed stone are not indicated.
3. the compositions of the concrete from which the aggregates are obtained are not entirely clear
4. The article provides an equation for the regression of concrete density on used aggregates. It is desirable to obtain models of other properties of concrete with recycled aggregates.
5. The work used a polycarboxylate superplasticizer at a certain dosage. The influence of the superplasticizer at other dosages is also of interest.
6. In conclusion, the authors extends the results obtained to all types of recycled aggregates. It should also be noted that they were obtained on the fillers that were studied in the article.
With minimal revision, the article can be published in the journal.
Author Response
The responses to the comments made by you after reviewing the paper can be found below:
1. The type of rock from which the recycled aggregates are obtained is unknown.
The concrete used to obtain the recycled aggregates does not come from a specific structure, so it is not possible to know the previous characteristics.
2. The strength characteristics of crushed stone are not indicated.
The type of rock from which the natural aggregates have been obtained is indicated. The strength characteristics of the original rock have not been analysed.
3. the compositions of the concrete from which the aggregates are obtained are not entirely clear
The concrete used to obtain the recycled aggregates does not come from a specific structure, so it is not possible to know the previous characteristics.
4. The article provides an equation for the regression of concrete density on used aggregates. It is desirable to obtain models of other properties of concrete with recycled aggregates.
The models obtained for the rest of the properties are more complex and suitable for evaluating the behaviour of the concrete. In any case, it is an interesting point to continue the line of research.
5. The work used a polycarboxylate superplasticizer at a certain dosage. The influence of the superplasticizer at other dosages is also of interest.
The proportion of admixture and the effective water/cement ratio are constant in all mixes to study whether the amount of recycled aggregate affects workability. However, the variation in the amount of admixture is of interest for future studies.
6. In conclusion, the authors extends the results obtained to all types of recycled aggregates. It should also be noted that they were obtained on the fillers that were studied in the article.